# An Assessment of Health Behaviours in Primary Care Patients: A Cross-Sectional Study

**DOI:** 10.3390/healthcare12141405

**Published:** 2024-07-15

**Authors:** Barbara Gawłowska, Ewelina Chawłowska

**Affiliations:** 1Department of Preventive Medicine, Poznan University of Medical Sciences, 60-781 Poznań, Poland; 2Laboratory of International Health, Department of Preventive Medicine, Poznan University of Medical Sciences, 60-781 Poznań, Poland

**Keywords:** healthy lifestyle, Health Behaviour Scale, primary health care, women, Poland, chronic diseases

## Abstract

A healthy lifestyle is a key determinant of health, especially for people with chronic diseases such as diabetes or hypertension. The health behaviours which contribute to a healthy lifestyle include the following: regular physical activity, preventive examinations, maintaining a proper diet and avoiding the consumption of alcohol or cigarettes. They have a significant impact on the disease process, controlling symptoms and preventing complications. The aim of this study was to examine the health behaviours related to diet, physical activity and prevention among Polish primary health care patients and to identify predictors of health behaviours. For this reason, the standardized Health Behaviour Scale (HBS) questionnaire was used. The study was conducted among 269 patients (including 61.71% women) in primary health care facilities. The mean age of participants was 52.89 years (±17.76). The mean HBS score was 36.93 ± 9.66. A statistically significant association was found between HBS scores and such sociodemographic variables as education (*p* = 0.0061, r = 0.17), body mass index (*p* = 0.0018, r = −0.20, β = −0.36) and self-assessed economic status (*p* = 0.0094, r = 0.16). Women’s health behaviours as measured by HBS were significantly better than men’s (*p* < 0.001, β = −6.82). A special focus should be given to the groups manifesting poorer health behaviours (e.g., men, older people and persons with low socioeconomic status) by offering them tailored health-promoting interventions.

## 1. Introduction

Health behaviours (health-related behaviours) according to the World Health Organization (WHO) refer to any actions that pertain to a person’s health and can impact their overall well-being, both in their day-to-day activities and in their utilization of health services. Those intentional or unintentional behaviours involve, for instance, participating in preventive interventions such as vaccination and cancer screening, following medical treatment plans, attending medical appointments, having good nutrition habits and getting enough physical activity and sleep. Risky behaviours such as consumption of alcohol and tobacco products are also identified as health-related behaviours. Health behaviours are often seen as actions taken by individuals, but they can be studied in and generalized to groups or entire populations. The behaviours, which largely contribute to people’s lifestyles, can change over time, in different stages of life, among different groups and in various environments [1,2,3,4,5,6].

There are a number of factors associated with the prevalence of particular health behaviours such as age, education level, economic status, health literacy and self-efficacy [7,8,9,10,11,12]. For example, in the Polish population, with age, the probability of engaging in physical activity decreases. On the other hand, wealthier Poles and those with higher education levels are more likely to be physically active [13].

Health behaviours influence health outcomes, but the association is by no means a simple one. There is a dynamic and multi-directional interplay between health behaviours, biological factors (e.g., genes), cognitive factors (e.g., decision-making) and social factors (e.g., cultural norms, food environment). The impact of particular factors varies across individuals [1,14,15,16,17,18,19,20,21]. Some behaviours, like physical activity or smoking, are more strongly associated with certain health outcomes than, for instance, diet [22]. Through a healthy diet or regular physical activity, individuals can prevent or delay the onset of conditions such as diabetes, hypertension and other cardiovascular diseases, chronic respiratory diseases or cancer [23].

Unfortunately, implementation of beneficial health behaviours is neither easy nor common [24], and this is true also about the Polish population. In two large-scale surveys conducted in 2003–2005 and 2013–2014, it was found that only 2% of Polish participants followed a healthy lifestyle, while 25% followed a poor lifestyle [25]. It is diet and tobacco smoking that are among the behavioural factors causing higher mortality, and their impacts differ between the sexes. Men are more likely to be exposed to risk factors such as smoking or excessive alcohol consumption. In 2022, the life expectancy for both sexes was 77.4 years (3.3 years below the European Union average), but it was 6 years shorter in men than in women [26]. As regards physical activity, approximately 64% of Poles over the age of 20 do not practice any. In the 40–59 age group, it is 58% of men and 66% of women who are not physically active [13].

What is more, similarly to other EU countries, a significant portion of the population in Poland is represented by elderly people. The percentage of people over 65 has increased from 12% in 2000 to 18% in 2020, and is expected to rise to 30% in 2050. Poland is among the EU countries with the highest prevalence of multiple chronic diseases in people over 65 years (50% of men and 60% of women). The prevalence of chronic diseases and disability is higher than the EU average [25].

The purpose of the following study was to assess health behaviours of primary care patients. Considering that primary health care is the patient’s first line of contact with health care, the role of primary healthcare professionals in guiding patients’ health behaviours is particularly important here. We also analysed relationships between certain health behaviours and demographic and socioeconomic variables. Identifying risk factors for harmful health behaviours and non-communicable diseases would help to develop relevant interventions. Moreover, the lack of research on the health behaviours of Poles and the crucial importance of primary health care made the following study necessary.

## 2. Materials and Methods

A cross-sectional study was conducted in Poland in the last quarter of 2023. Data were collected among adult patients in primary health care facilities in Poznań (Poland) through convenience sampling. The inclusion criteria for the study were persons aged over 18 years old, the status of primary care patient, good command of the Polish language and willingness to participate in the study. It was a paper-and-pencil survey, which was filled by patients themselves in waiting rooms of primary care clinics before medical appointments. The interviewer personally invited people to participate in the survey. Of those invited, 32% agreed to participate. Participation in the survey was fully anonymous and voluntary. Each respondent had been informed of the purpose of the study and the voluntary and anonymous nature of participation. The responses were collected only from those respondents who had provided written consent to participate in the study.

The Bioethics Committee of the Poznan University of Medical Sciences confirmed that the study did not constitute a scientific experiment and, as a result, did not need ethical approval in accordance with the Polish law (decision KB-902/23).

The questionnaire used in this study was a validated Health Behaviour Scale (HBS) [27]. It included 22 items, which were categorized into 5 domains. The domains focused on preventive behaviours related to the healthcare system (3 items), individual preventive behaviours (6 items), health behaviours related to diet (7 items), health behaviours related to physical activity (3 items) and unhealthy behaviours (3 items). Questions included performing such examinations as: cholesterol, blood sugar, blood pressure, breast self-examination and smear test. Questions on risky behaviours considered smoking (including e-cigarettes) and alcohol consumption. Behaviours related to mental health (stress management, time for rest and sleep) were also addressed. The items were scored on a scale from 0 to 3 points. The higher values represented more beneficial health behaviours. The maximum possible total score (denoting optimum health behaviours) was 60 for men and 66 for women. All demographic data such as sex, age, education, economic status, residence, height, weight and health conditions were self-reported by participants.

Data deficiencies were occasionally present. Only when a participant answered all questions in a particular HBS domain, was the domain included in statistical analyses. PQStat v.1.8.6 was used to perform statistical analysis. To evaluate data normality, a Shapiro–Wilk test was used. The relationship between variables was assessed by the Spearman correlation coefficient and a Mann–Whitney U test. The significance level of *p* = 0.05 was adopted in all analyses.

The authors investigated relationships between health behaviours and explanatory variables in multivariate models. Linear regression models were performed for the following continuous variables: total HBS score, preventive behaviours related to the healthcare system, individual preventive behaviours, health behaviours related to diet, health behaviours related to physical activity and unhealthy behaviours. The following explanatory categorical variables were introduced into the models: sex (categorised into “male”—a reference category, and “female”), residence (categorised into “village”—a reference category, “city up to 50,000 inhabitants”, “city above 50,000 inhabitants” and “city above 200,000 inhabitants”), education (categorised into “primary + vocational”—a reference category, “secondary” and “higher”), economic status (categorized into “very bad”—a reference category, “bad”, “average”, “good” and “very good”), having diabetes (categorized into “No”—a reference category, and “Yes”) and having hypertension (categorized into “No”—a reference category, and “Yes”). The continuous variables used in models were age, BMI and the number of health conditions.

## 3. Results

### 3.1. Population Characteristics

Table 1 presents population characteristics. The study was conducted among 269 patients including 61.71% women. The mean age of respondents was 52.89 years (±17.76). Over 67% of the study group was from a city with above 200,000 inhabitants. A majority of the group were patients with good economic status (46.46%) and secondary education (44.19%). The mean body mass index (BMI; calculated on the basis of self-reported weight and height) was 26.19 (±4.81). More than half of the respondents (51.80%) were overweight or obese (according to the WHO criteria [28]) and nearly 65.5% claimed at least one chronic disease. The most common diseases were hypertension (37.18%) and diabetes (14.87%).

### 3.2. HBS Domains

Table 2 presents HBS results. The mean HBS score obtained by respondents was 36.90 ± 9.81. The maximum score possible was 66. Out of the five domains, domain 1 (Preventive behaviours related to healthcare system), which included screening exams, had the lowest mean score (1.51 ± 0.77). The patients obtained the highest score in Unhealthy behaviours (2.23 ± 0.59), which indicated that the study group relatively rarely used alcohol and tobacco products.

Table 3 shows the distribution of patients’ engagement in health behaviours related to diet and physical activity. Approximately 63% of the study group responded “yes” or “rather yes” when asked if they led active lifestyles. Regarding behaviours related to diet, over 14% of the patients limited salt and sugar consumption, nearly 71% ate breakfast, but only 10.45% ate 4–5 portions of fruit and vegetables every day.

### 3.3. Associations between Health Behaviours and Sociodemographic Variables

Correlations between total HBS and HBS domains with dichotomous variables such as sex, having hypertension and diabetes were performed using the Mann–Whitney U test (Table 4). The Spearman correlation coefficient was used to examine associations between health behaviours with BMI, age, education, residence, economic status and number of health conditions (Table 5). The analyses showed significant positive associations between health behaviours and sociodemographic variables such as sex (*p* < 0.001), education level (*p* = 0.0061), economic status (*p* = 0.0094) and significant negative association with BMI (*p* = 0.0018). Female respondents had better health behaviours in comparison to men. The total questionnaire score increased in a direct proportion to education level and decreased in an inverse proportion to BMI. Patients with several chronic diseases more often performed preventive examinations (M = 1.71) than people with no health conditions (M = 1.29). They were also more likely to smoke cigarettes and consume alcohol (M = 2.01) but exercise less often (M = 1.26). People with a better self-assessment of their financial situation led more active lifestyles and had better nutrition habits and preventive behaviours. Patients over the age of 55 performed preventive examinations more frequently. Respondents aged 40–55 were the least likely to smoke cigarettes and consume alcohol.

### 3.4. Multivariate Analysis

Appendix A present linear regression analysis. A regression analysis for total HBS score (Appendix A) found sex (*p* < 0.001) and BMI (*p* = 0.0198) to be significant predictors of general health behaviours. Women had significantly higher HBS scores (β = 6.82) compared to men, and so did patients with lower BMIs (β = −0.36) (Appendix A). Sex was also a positive predictor (with males used as a reference group) for health behaviours related to diet (β = 0.23). BMI was identified as a statistically significant negative predictor for health behaviours related to physical activity (β = −0.06) and unhealthy behaviours (β = −0.03). A higher education level significantly predicted a better score in the unhealthy behaviours domain as well (β = 0.30) (Appendix A). There was no significant correlation between the explanatory variables and the preventive behaviours related to healthcare system domain.

### 3.5. Health Behaviours among Women

In the present study, correlations between health behaviours and sex were found. Women had significantly higher HBS scores in comparison to men. Table 6 presents an analysis of the HBS factors validated in a 2022 study among women [27]. Out of the four factors, women obtained the highest score in Diet and Mental Health (1.91 ± 0.55). It indicates that female respondents maintained a healthy varied diet by limiting salt and sugar consumption. In addition, they usually found time to relax, had enough sleep and were able to manage stress as well. However, the women scored the lowest on Individual Healthy Behaviours (1.64 ± 0.72). This factor includes looking for information on healthy eating, checking one’s body for physical lesions, performing a breast self-examination and checking the composition of food products.

A statistically significant association was identified between factors and demographic variables (Table 7). Overweight and obese females reported checking cholesterol and blood sugar levels more often compared to women with normal BMIs (*p* = 0.0023). They were also less likely to engage in physical activity (*p* < 0.001) and individual healthy behaviours (*p* = 0.0054). Chronic diseases like hypertension or diabetes correlated with a less active lifestyle, more frequent participation in preventive examinations and less frequent individual behaviours. Older female respondents had significantly better behaviours linked to preventive examinations (*p* = 0.0024), diet and mental health (*p* = 0.001). However, they obtained lower scores in Individual Healthy Behaviours than younger females (*p* = 0.0039).

## 4. Discussion

The present study showed correlations of certain health-related behaviours with health outcomes. For instance, physical activity behaviours and risky behaviours were associated with BMI, number of health conditions and having hypertension and diabetes. Respondents without any chronic diseases and with a normal BMI had significantly better health behaviours related to physical activity than the rest of the patients. Poorer nutrition habits were more often identified in respondents with several chronic diseases. Patients with health conditions such as hypertension or diabetes were more likely to smoke and drink alcohol.

In addition, our findings showed associations between particular health behaviours and sociodemographic variables. Older participants of the study had lower physical activity levels. To compare, in a study conducted in 2022, authors examined the relationship between several variables and physical activity among 72,262 middle-aged and older individuals in India. It was found that nearly 40% of participants did not have an adequate level of physical activity, with a lower prevalence among older age groups. Participants with chronic conditions such as hypertension, heart disease, stroke and diabetes were also found to be less physically active, similarly to the observations in the present study [29]. Also, older participants of an American analytical study from 2010 were consistently less active [30]. Another association found in our respondents was that, between physical activity and sex, men tended to be less physically active, similarly to participants of a number of other studies [29,31,32], although there is evidence indicating that the difference might be attributed to the generally poorer health status of men [30]. Finally, physical activity levels are oftentimes linked to higher education levels. For instance, in a 2016 study, young women with secondary or higher education more often exhibited adequate physical activity behaviours than women with a vocational education (*p* = 0.010) [33]. A similar association was observed in our respondents, even though the study group was generally older.

As regards nutrition, the present study also showed that over 70% of primary care patients ate breakfasts, 10.45% ate fruits and vegetables daily and about 14% limited sugar and salt consumption. Female participants had significantly better nutrition habits than male participants. In a 2012 study of 567 Polish teachers, authors used the Positive Health Behaviour Scale (PHBS) to assess nutrition-related behaviours. The study found that 57% of teachers reported eating breakfast regularly, while 35.2% ate fruit daily and 36.4% ate vegetables daily. However, only 12.5% of participants reported limiting their consumption of sweets. The study also revealed that male teachers exhibited poorer dietary behaviours compared to their female counterparts [31,32]. Katarzyna Hildt-Ciupińska and Karolina Pawłowska-Cyprysiak in 2020 presented results of a health behaviours survey in 606 Polish men. Approximately 37% of men reported eating breakfasts, 27% eating fruits, 25% eating vegetables and 15% limiting salt and sweets consumption [34]. In another Polish study using PHBS among 200 Polish women aged 18–35, a significant association of nutrition behaviours with age (*p* < 0.001) and education level (*p* < 0.001) was identified [28]. Our respondents represented a different age group, and we were not able to associate better nutrition with age. However, we did find such an association in relation to higher levels of education.

Factors associated with preventive health behaviours were also explored in numerous studies [35,36,37,38]. In general, women tend to exhibit them more regularly than men. It was observed in students [38] as well as in general population [35,36,37]. The frequency of such behaviours is often associated with age. For example, in postmenopausal women, preventive behaviours, as measured with the Health Behaviour Inventory (HBI), were the most prevalent of all health-related behaviours [39,40], but the least prevalent in a younger study group of pregnant and non-pregnant women [41]. In a 2009 Australian study, older people of both sexes (>51 years old) more frequently performed preventive examinations and individual examinations like checking blood pressure [42]. The association between age was also observed in our respondents with respect to preventive behaviours related to the healthcare system. However, the prevalence of particular preventive behaviours may differ widely. For example, the Polish female teachers surveyed in 2012 rarely performed such preventive tests as breast self-examination (14.5%) [31], just as our primary care patients (22.89%). The use of testicular self-examination was similarly low (14.8%) among the male teachers [31]. On the other hand, 47% of men participating in the 2020 study reported taking part in prophylactic examinations recommended by physicians [34], compared to 81.03% of our respondents checking their cholesterol levels and 89.23% of those checking glycemia.

The relationships between behaviours related to mental health and different socioeconomic factors may be quite complex. In a group of young non-pregnant women, such behaviours were the most prevalent of all health-related behaviours [41], but rather rare among students of health professions [38], indicating the importance of environmental demands. Mental health behaviours may also be associated with economic status; in a 2016 survey among women of reproductive age, better socioeconomic status was linked to better behaviours related to mental health [33]. Also in our survey, markedly poorer behaviours related to sleep, relaxing and stress management were found in persons with lower self-reported economic statuses.

When health-related behaviours are assessed as a whole, certain socioeconomic variables may also play a role. For example, in a study among Polish men, better material status was linked to significantly higher results in a Health Behaviour Scale [34], just as in the present study. Gender may also significantly differentiate general health behaviours, as it turned out in the present findings. Similarly, women’s overall health behaviours were found to be better than men’s in studies among students [38,43] as well as in the general population [44,45]. Such tendencies are not universal, but whenever they are identified in a population, they should be taken into account while planning public health interventions.

It should be noted that due to convenience sampling, the present study group is not representative of primary care patients in Poland. Therefore, it would be recommended to carry out larger-scale research devoted to health behaviours in this health setting. The current study succeeded in including primary care patients, which is beneficial for the representativeness of the population. In fact, primary health care patients represent all social groups with a variety of health problems. Primary health care is the most accessible level of health care and a multicentre study is recommended in the future.

## 5. Conclusions

Health-related behaviours are strongly related to the health status of the individual and the population. Therefore, it is important to learn about these behaviours in various populations and the factors contributing to their levels. It may enable mapping health needs and making changes to improve the health situation. However, improving health behaviours among adults is challenging because of difficulty getting through to this population. Therefore, it would be advisable to develop opportunistic interventions for primary health care, where a significant portion of the population can be reached. The present results suggest that future educational and preventive programmes delivered within primary healthcare could focus on neglected populations including men, elderly people and persons with low socioeconomic status.

Thus, the current study may have important implications for further research and health interventions. With such data collected, medical personnel can be better informed in the area of patients’ needs and challenges. It would allow them to effectively provide health education and advise patients on improving their habits. In addition, the results of the survey can be used to compare health behaviours of Polish patients with those of patients in other countries, which may be important for developing standardized health strategies.

## Figures and Tables

**Table 1 healthcare-12-01405-t001:** Demographic characteristics.

n = 269		n	%
Sex	Female	166	61.71
	Male	103	38.29
Age (n = 252)	≤39	66	20.91
	40–55	67	30.07
	56–69	63	27.45
	≥70	57	21.57
Residence (n = 267)	Village	48	17.98
	City up to 50,000 inhabitants	31	11.61
	City above 50,000 inhabitants	9	3.37
	City above 200,000 inhabitants	179	67.04
Education (n = 267)	Primary (+vocational)	42	15.73
	Secondary	118	44.19
	Higher	107	40.07
Economic status (n = 254)	Very bad	4	1.57
	Bad	13	5.12
	Average	96	37.80
	Good	118	46.46
	Very good	27	10.63
BMI (n = 249)	<18.50 (Underweight)	4	1.61
	18.50–24.99 (Normal)	116	46.59
	25.00–29.99 (Overweight)	84	33.73
	≥30.00 (Obesity)	45	18.07
Number of health conditions	0	93	34.57
	1	81	30.11
	2	57	21.19
	≥3	38	14.13
Hypertension	No	169	62.83
	Yes	100	37.18
Diabetes	No	229	85.13
	Yes	40	14.87

**Table 2 healthcare-12-01405-t002:** Domains of Health Behaviour Scale.

DOMAINS	Mean	SD	95% CI	Median
Preventive behaviours related to healthcare system	1.51	0.77	1.42–1.61	1.50
Individual preventive behaviours	1.70	0.56	1.63–1.77	1.80
Health behaviours related to diet	1.75	0.61	1.68–1.82	1.86
Health behaviours related to physical activity	1.58	0.83	1.48–1.68	1.67
Unhealthy behaviours	2.23	0.59	2.16–2.30	2.33
**Total HBS**	36.90	9.81	35.73–38.08	37.00

SD—standard deviation; CI—confidence interval.

**Table 3 healthcare-12-01405-t003:** HBS items related to diet and physical activity.

Item	Yes % (n)	Rather Yes % (n)	Rather No % (n)	No % (n)
1. I lead an active lifestyle	22.26 (59)	40.76 (108)	27.55 (73)	9.43 (25)
2. My diet is varied	23.88 (64)	52.61 (141)	18.28 (49)	5.22 (14)
4. I limit the consumption of sugar and foods which contain it (sweets)	14.18 (38)	39.18 (105)	33.58 (90)	13.06 (35)
6. I look for information on healthy eating	20.97 (56)	37.83 (101)	25.09 (67)	16.11 (43)
7. I limit the consumption of salt and foods which contain it	13.43 (36)	40.30 (108)	34.33 (92)	11.94 (32)
10. I eat breakfast	70.90 (190)	19.78 (53)	5.22 (14)	4.10 (11)
16. I use daily activities as an opportunity for physical activity (e.g., I climb the stairs instead of using the elevator, park my car at a distance so that I can walk, I move around by bicycle)	30.60 (82)	39.93 (107)	18.28 (49)	11.19 (30)
17. When buying food products, I check their composition	21.43 (57)	37.59 (100)	24.81 (66)	16.17 (43)
22. I eat 4–5 portions of fruit and vegetables per day	10.45 (28)	34.70 (93)	39.18 (105)	15.67 (42)
	**at least 3 times a week**	**1–2 times a week**	**several times a month**	**never**
19. I regularly perform physical exercise	13.81 (37)	20.52 (55)	29.48 (79)	36.19 (97)

**Table 4 healthcare-12-01405-t004:** Correlations between health behaviours and sociodemographic variables with the use of Mann–Whitney U test.

	Preventive Behaviours Related to Healthcare System M ± SD	Individual Preventive Behaviours M ± SD	Health Behaviours Related to Diet M ± SD	Health Behaviours Related to Physical Activity M ± SD	Unhealthy Behaviours M ± SD	Total HBS M ± SD
**Sex** Female Male	*p* = 0.0615 d = 0.2294 1.58 ± 0.66 1.4 ± 0.91	*p* = 0.9378 d = 0.0096 1.7 ± 0.57 1.71 ± 0.54	***p* < 0.001** **d = 0.4449** 1.86 ± 0.59 1.57 ± 0.59	*p* = 0.1736 d = 0.1666 1.64 ± 0.76 1.48 ± 0.91	***p* = 0.0047** **d = 0.3487** 2.33 ± 0.51 2.08 ± 0.66	***p* < 0.001** **d = 0.8526** 39.87 ± 9.33 32.13 ± 8.64
**Hypertension** No Yes	***p* = 0.0133** **d = 0.3047** 1.43 ± 0.77 1.65 ± 0.75	***p* = 0.0324** **d = 0.2628** 1.76 ± 0.56 1.6 ± 0.55	*p* = 0.2109 d = 0.1531 1.79 ± 0.62 1.67 ± 0.57	***p* < 0.001** **d = 0.4606** 1.74 ± 0.8 1.32 ± 0.81	***p* = 0.0367** **d = 0.2566** 2.31 ± 0.51 2.11 ± 0.68	*p* = 0.4008 d = 0.1027 37.47 ± 9.93 35.95 ± 9.56
**Diabetes** No Yes	***p* = 0.0538** **d = 0.2366** 1.47 ± 0.78 1.74 ± 0.68	*p* = 0.6557 d = 0.0546 1.7 ± 0.57 1.68 ± 0.54	*p* = 0.1545 d = 0.1745 1.77 ± 0.58 1.61 ± 0.72	***p* = 0.001** **d = 0.4047** 1.65 ± 0.8 1.16 ± 0.86	***p* < 0.001** **d = 0.4378** 2.28 ± 0.57 1.96 ± 0.6	*p* = 0.1553 d = 0.1742 37.29 ± 9.69 34.7 ± 10.29

M ± SD—mean ± standard deviation.

**Table 5 healthcare-12-01405-t005:** Correlations between health behaviours and sociodemographic variables with the use of Spearman correlation coefficient.

	Preventive Behaviours Related to Healthcare System M ± SD	Individual Preventive Behaviours M ± SD	Health Behaviours Related to Diet M ± SD	Health Behaviours Related to Physical Activity M ± SD	Unhealthy Behaviours M ± SD	Total HBS M ± SD
**Age** ≤39 40–55 56–69 ≥70	***p* < 0.001** **r = 0.2154** 1.19 ± 0.72 1.53 ± 0.73 1.76 ± 0.83 1.6 ± 0.7	*p* = 0.4570 r = 0.0471 1.67 ± 0.56 1.61 ± 0.61 1.79 ± 0.62 1.72 ± 0.45	*p* = 0.4797 r = −0.0447 1.71 ± 0.55 1.86 ± 0.62 1.82 ± 0.65 1.59 ± 0.63	***p* < 0.001** **r = −0.2265** 1.82 ± 0.74 1.75 ± 0.65 1.46 ± 0.94 1.29 ± 0.9	***p* = 0.0043** **r = −0.1795** 2.28 ± 0.56 2.46 ± 0.39 2.14 ± 0.6 2.01 ± 0.74	*p* = 0.8887 r = 0.0089 35.56 ± 8.43 38.4 ± 9.83 37.98 ± 11.2 35.5 ± 10.08
**Education** Primary or vocational Secondary Higher	*p* = 0.3521 r = 0.0572 1.54 ± 0.78 1.4 ± 0.82 1.62 ± 0.7	*p* = 0.5461 r = 0.0371 1.7 ± 0.6 1.7 ± 0.52 1.71 ± 0.6	***p* < 0.001** **r = 0.2049** 1.51 ± 0.55 1.71 ± 0.65 1.88 ± 0.55	***p* = 0.0024** **r = 0.1847** 1.25 ± 0.89 1.56 ± 0.86 1.73 ± 0.73	***p* = 0.0046** **r = 0.1730** 1.98 ± 0.69 2.21 ± 0.63 2.35 ± 0.46	***p* = 0.0061** **r = 0.1674** 34.12 ± 9.55 36.14 ± 10.48 38.75 ± 8.89
**Economic status** Very bad Bad Average Good Very good	***p* = 0.0051** **r = 0.1738** 0.46 ± 0.42 1.15 ± 0.66 1.4 ± 0.76 1.64 ± 0.73 1.51 ± 0.84	***p* = 0.0205** **r = 0.1443** 1.28 ± 0.46 1.45 ± 0.57 1.66 ± 0.55 1.78 ± 0.57 1.74 ± 0.57	***p* = 0.0184** **r = 0.1467** 1.36 ± 0.7 1.33 ± 0.42 1.72 ± 0.59 1.82 ± 0.63 1.79 ± 0.6	***p* = 0.0418** **r = 0.1268** 1.92 ± 1.07 0.87 ± 0.87 1.55 ± 0.77 1.64 ± 0.84 1.73 ± 0.83	*p* = 0.9896 r = −0.0008 2.67 ± 0.27 1.85 ± 0.85 2.24 ± 0.59 2.26 ± 0.57 2.15 ± 0.56	***p* = 0.0094** **r = 0.1615** 29.25 ± 9.74 30.38 ± 8.02 36.13 ± 9.58 38.44 ± 9.94 37.3 ± 10.25
**BMI** Underweight Normal Overweight Obesity	*p* = 0.5340 r = 0.0396 1 ± 0.72 1.45 ± 0.72 1.77 ± 0.79 1.26 ± 0.81	*p* = 0.2878 r = 0.0676 1.9 ± 0.35 1.74 ± 0.59 1.7 ± 0.61 1.65 ± 0.44	***p* = 0.0007** **r = −0.2123** 1.96 ± 0.62 1.85 ± 0.61 1.81 ± 0.56 1.46 ± 0.6	***p* < 0.001** **r = −0.3582** 1.58 ± 0.79 1.88 ± 0.81 1.58 ± 0.7 0.94 ± 0.76	***p* = 0.0006** **r = −0.2164** 2.42 ± 0.5 2.31 ± 0.56 2.33 ± 0.44 1.85 ± 0.71	***p* = 0.0018** **r = −0.1973** 39.75 ± 9.14 38.33 ± 10.04 38.24 ± 9.2 31.91 ± 9.4
**Number of health conditions** 0 1 2 ≥3	***p* = 0.0004** **r = 0.2165** 1.29 ± 0.82 1.56 ± 0.69 1.68 ± 0.77 1.71 ± 0.67	*p* = 0.2937 r = −0.0643 1.77 ± 0.59 1.66 ± 0.57 1.62 ± 0.54 1.73 ± 0.47	*p* = 0.0529 r = −0.1181 1.85 ± 0.62 1.73 ± 0.57 1.7 ± 0.57 1.61 ± 0.71	***p* < 0.001** **r = −0.2534** 1.85 ± 0.74 1.59 ± 0.76 1.34 ± 0.79 1.26 ± 1	***p* = 0.0059** **r = −0.1644** 2.39 ± 0.46 2.18 ± 0.54 2.2 ± 0.66 2.01 ± 0.72	*p* = 0.6411 r = −0.0285 37.56 ± 10 36.67 ± 9.36 36.7 ± 10.04 36.11 ± 10.21

M ± SD—mean ± standard deviation.

**Table 6 healthcare-12-01405-t006:** Validated factors of Health Behaviour Scale in women.

FACTORS (n = 166)	Mean	Median	SD	95% CI
F1: Diet and Mental Health	1.91	2.00	0.55	1.83–2.00
F2: Individual Healthy Behaviours	1.64	1.75	0.72	1.53–1.75
F3: Preventive Behaviours	1.67	2.00	0.78	1.55–1.79
F4: Physical Activity	1.87	2.00	0.78	1.75–2.00

SD—standard deviation, CI—confidence interval.

**Table 7 healthcare-12-01405-t007:** Correlations between factors and sociodemographic variables in women.

	F1: Diet and Mental Health M ± SD	F2: Individual Healthy Behaviours M ± SD	F3: Preventive Behaviours M ± SD	F4: Physical Activity M ± SD
**BMI** Underweight (n = 4) Normal (n = 81) Overweight (n = 46) Obesity (n = 23)	*p* = 0.3913 r = −0.0696 2 ± 0.49 1.95 ± 0.58 1.9 ± 0.56 1.93 ± 0.36	***p* = 0.0054** **r = −0.2230** 2 ± 0.35 1.72 ± 0.79 1.57 ± 0.63 1.43 ± 0.75	***p* = 0.0023** **r = 0.2435** 1.13 ± 1.03 1.48 ± 0.76 2.03 ± 0.67 1.67 ± 0.85	***p* = 0.0001** **r = −0.3126** 1.63 ± 0.95 2.07 ± 0.74 1.82 ± 0.69 1.28 ± 0.85
**Number of health conditions** 0 (n = 54) 1 (n = 52) 2 (n = 37) 3≤ (n = 23)	*p* = 0.7678 r = 0.0231 1.99 ± 0.55 1.82 ± 0.51 1.88 ± 0.58 2 ± 0.56	***p* = 0.0163** **r = −0.1863** 1.85 ± 0.66 1.58 ± 0.75 1.48 ± 0.73 1.58 ± 0.7	***p* = 0.0010** **r = 0.2522** 1.46 ± 0.77 1.62 ± 0.73 1.81 ± 0.83 2.04 ± 0.69	***p* = 0.0003** **r = −0.2776** 2.2 ± 0.59 1.85 ± 0.65 1.61 ± 0.83 1.59 ± 1.04
**Hypertension** Yes (n = 56) No (n = 110)	*p* = 0.7651 d = 0.0466 1.87 ± 0.56 1.93 ± 0.54	***p* = 0.0017** **d = 0.4970** 1.41 ± 0.7 1.76 ± 0.71	***p* = 0.0040** **d = 0.4562** 1.9 ± 0.75 1.55 ± 0.77	***p* = 0.0007** **d = 0.5381** 1.56 ± 0.81 2.03 ± 0.71
**Diabetes** Yes (n = 21) No (n = 145)	*p* = 0.2518 d = 0.1794 2.03 ± 0.52 1.89 ± 0.55	***p* = 0.1344** **d = 0.2344** 1.45 ± 0.72 1.67 ± 0.72	***p* = 0.0180** **d = 0.3715** 2.05 ± 0.65 1.61 ± 0.78	***p* = 0.0060** **d = 0.4315** 1.4 ± 0.83 1.94 ± 0.75

M ± SD—mean ± standard deviation.

## Data Availability

The datasets generated and analysed during the current study are available from the corresponding authors.

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
