# Peer review of "An Assessment of Health Behaviours in Primary Care Patients: A Cross-Sectional Study"

_healthcare, 2024, doi:10.3390/healthcare12141405_

Round 1
Reviewer 1 Report
Comments and Suggestions for Authors
The study conducts a short health behavior scale in a small primary care patient convenience sample. The manuscript falls short of articulated a strong rationale for conducting the study, given the small sample size and simplistic methods. There are various issues throughout the manuscript that need to be addressed as below.
Abstract
Consider amending sentence two for enhanced clarity
Keywords: use keywords that are relevant and don’t already feature in the title
Introduction
Line 34. Amend to physical activity “and” sleep, rather than “or”
The introduction fails to describe the relevance or need to conduct this study. Further discussion needs to be added to articulate this. There should be a clear rationale for the target population and the value of findings to the existing literature base.
Methods
can you describe why ethical approval was not required? Since data was collected on human participants, ethics approval is a standard requirement prior to collecting data in most countries.
Did participants return an informed consent form prior to participating in data collection?
How was participation advertised? What proportion of potential participants volunteered to participate?
What were the procedures, how and when was the questionnaire completed during their visit.
The methodology is very concise and lacks descriptive depth, please re-address this section with comprehensive detail on methodology used and all data analysis procedures. Was data normality evaluated to determine the type of data analysis techniques used?
How was missing data addressed?
Results
what is deemed good socioeconomic status? Such classifications need to be described in detail in the methods
Overweight and obesity… again what criteria is this based on?
Do higher values of HBS denote better or worse health? Another detail needed in methods
Discussion
Enhanced discussion of the relevance of the present findings and their value to the scientific body is needed
You need to add comprehensive discussion of the limitations of your study
Comments on the Quality of English LanguageComplete a general grammar and spelling check throughout
Author Response
Dear Reviewer,
Please find the attached file with our responses.
Yours faithfully,
The authors

Reviewer 2 Report
Comments and Suggestions for Authors
I appreciate the opportunity to review the referenced manuscript and commend the authors for their exemplary work performed. Health behaviors are the cornerstone for healthspan and lifespan. As such, evaluating the relationship between health behaviors and exploratory variables, as well as for comparisons between the sexes is of high interest to the world of precision medicine. However, there are some concerns that require addressing prior to recommendation for publication.
TITLE
The title is misleading. The authors pose the question of "do women lead healthy lifestyles?" leading the reader to believe this will be a primary assessment of health behaviors in women and disentangling factors that may affect health behaviors in women. Yet, that is not the case and I recommend a title that more appropriately reflects the content of the manuscript.
ABSTRACT
Has the Health Behavior Scale (HBS) questionnaire been validated, specifically the short-form version? In this vein, is the HBS appropriate for the Polish population? This would resonate extremely well if placed in the abstract, if so. If not, the recommendation is to note that this is a non-validated questionnaire, but captures each respective health behavior of interest, justifying its use in the study.
Would suggest at least putting the mean+-SD age and proportion of participants that identified as male/female.
Another suggestion is discerning the direction of the relationship (i.e., positive or negative). Need to define BMI prior to acronym use. Also need to identify is certain variables were evaluated as continuous or categorical. Other than the p value, if you would like to cut-out directionality, you could place the correlation coefficient or beta estimate is using regression analyses, as appropriate. Last suggestion would be to highlight the score differences between men and women and not the effect size. A large effect size may be due to sample size and distribution of the data and not necessarily the degree of difference between the sexes, so the findings may be lost.
INTRODUCTION
This is more nitpicky than anything, but using an acronym for health behaviors feels unnecessary. Ultimately the authors call, but I would spell out health behaviors rather than using HB acronym.
Another nitpicky comment is consistency in writing. For example, preventive versus preventative. I once heard a talk by Steven Blair, the father of physical activity epidemiology, who vehemently discouraged the use of the word "preventative". I would suggest using preventive throughout.
The language structure of the paragraphs make the manuscript a bit difficult to read. Proper punctuation and transition statements.
Back to the consistency bit, but health behaviors and lifestyle are being used interchangeably. Lifestyle, per your provided definition of health behaviors, falls under your definition of a health behavior, but is not equivalent to health behavior.
The introduction is lacking substance. There needs to be some more built out justification. The second paragraph should be dedicated to health behaviors and their impact on health outcomes. The third paragraph should discuss the determinants of health behaviors from prior published studies. There could be a third paragraph dedicated to the Polish population and how this has never been evaluated to this extent using this target demographic. For example, some data to support how many adults in the Polish population engage in physical activity or proper nutritional habits would be extremely helpful. What the demographics of the general population are, so on and so forth. The author needs to better set the stage for why all of this is important and what the findings will value add to discern importance of the project.
MATERIALS AND METHODS
The Materials and Methods section is severely lacking.
My thoughts in the abstract are of greater importance here.
Were there any sample size estimations performed? I am not a believer in that you need power calculations, but something the discern the authors had a presumptive sample size they wished to achieve. For instance, how many unique patients frequent the primary healthcare facilities in Poznan (Poland)? Of those, how many were approached with the survey compared to how many completed? Did the surveys need to be complete for inclusion into analyses or if they answered all question specific to certain domains only those domains were included?
There needs to be discussion about the demographic data captured, how it was captured, and what type of questions were asked, as examples.
The authors now state that the HBS is validated. Is that the full-length form or the utilized short-form? Is the HBS validated for Polish people. Was there an overall composite score for HBS, which is further broken down into the 5 domains? Although the authors reference the questionnaire, there is value in providing the questionnaire as a supplement or inputting the questions in the materials and methods section for ease of access. Without ease of review of this information, it is difficult to understand what type of questions led into the overall domain.
The authors need to identify which variables were assessed as categorical and/or continuous, then identify the statistical tests performed to each variable type.
RESULTS
Individual preference, but cutting down to one decimal place would add clarity.
It appears that the authors "cherry picked" the questions they wished to highlight in Table 3. In the materials and methods, there are 7 items related to diet and 3 items related to physical activity. To avoid a comment I made earlier about the materials and methods expanding on the HBS, the authors could instead, expand out this section in the results and add in all data from each question of the HBS.
Table 4 is the colloquial money maker. My comment is that there needs to be a delineation between what is a comparison and p value for the relationship using continuous variable analyses versus categorical. Setting a reference group for the categorical analysis and then having a beta estimate as to what a group increase or decrease would be would add more value as it would allow for an estimation to be evaluated in the real-world rather than a degree of difference from the sampled population, if that makes sense.
There is not justification mentioned for why the authors specifically chose to further examine the health behaviors in women specifically, when there were also significant relationships with other factors. The authors need to justify why single out sex as a focal point, add in the modeling for men, and then utilize sex as an interaction term to identify is women had better health behaviors compared to men and if the relationship with other factors continued to stand true.
DISCUSSION
The discussion section is fine for now, but will need to be revised based on author corrections following review and re-evaluated by the reviewers again.
CONCLUSION
There needs to be better conclusions. Why would you want your last take-home message to be a limitation? Why not instead have a core strengths and limitations section? Then focus on what the next steps are.
Your conclusion section should set the stage for what is next to come and why this work was important to inform these next steps, not simply re-highlight the findings and present further challenges and future considerations.
Comments on the Quality of English LanguageMinor editing required.
Author Response

(The authors gave the same response as above.)

Reviewer 3 Report
Comments and Suggestions for Authors
Once my feedback is included, the paper will be fit for publication.

Comments on the Quality of English LanguageThe English needs some profreading.
Author Response

(The authors gave the same response as above.)

Round 2
Reviewer 1 Report
Comments and Suggestions for Authors
The authors have addressed most of the comments from the previous review. The conclusions section can still be improved. The discussion of study limitations should be placed at the end of the discussion section instead of in the conclusions section.
Comments on the Quality of English LanguageGeneral spelling and grammar check is advised.
